# Cell sorting based on pulse shapes from angle resolved detection of scattered light
Daniel Kage [1], Andrej Eirich[2], Kerstin Heinrich[1], Jenny Kirsch[1], Jan Popien[2], Alexander Wolf[1], Konrad v. Volkmann[2], Hyun-Dong Chang [1,3] & Toralf Kaiser [1] ✉

Flow cytometry is a key technology for the analysis and sorting of cells or particles at high throughput. Conventional and current flow cytometry is primarily based on fluorescent stains to detect the cells of interest. However, such stains also have disadvantages, as their effect on cells must be carefully tested to avoid effects on the results of the experiments. Alternative approaches using imaging or other label-free techniques often require highly sophisticated setups, are commonly limited in resolution, and produce challenging amounts of data. Our technology exploits scattered light instead. The custom-built flow cytometry setup comprises a fiber array in forward scatter detection for angular resolution and captures the whole pulse shape with advanced signal processing. Thereby this setup enables cell analysis and sorting purely based on scattered light signals without the need for fluorescent labels. We demonstrate the feasibility of this cell sorting technology by sorting cell lines for their cell cycle stages based on scattered light. Furthermore, we demonstrate the ability to classify human peripheral blood T- and B-cell subsets.

Analyzing and sorting cells at high throughput rates is a cornerstone in modern bio-medical research. A key technology for this purpose is flow cytometry, in particular fluorescence-activated cell sorting which enables high-speed cell sorting by distinguishing cell characteristics predominantly based on fluorescent markers and basic light scatter analysis. Recent advances in flow cytometry have made it possible to differentiate human peripheral blood immunophenotypes using more than 40 different fluorochromes with full spectrum detection[1,2]. However, their potential toxicity requires careful consideration and optimization to avoid adverse effects on cellular functions and experimental outcomes[3]. In addition, the staining process is becoming increasingly complex. This places high demands on sample preparation to avoid errors[4].

The cellular morphology has long been neglected as a parameter for high-speed sorting approaches, even though it can provide valuable information about cells. Cell morphology can reflect genomic heterogeneity and is related to cellular functionality, and may therefore be an important indicator of, for instance, tumorigenic and metastatic potential[5,6], or functional properties of cells[7]. In order to make cell morphology accessible at high throughput in flow cytometry, imaging flow cytometers[8,9] with sorting capabilities were developed recently. Furthermore, an approach based on structured illumination ready for cell sorting using AI data analysis was introduced[10,11]. Although this approach does not use image reconstruction, it is constrained by the same limitations as other image-based methods[10] and has relatively low sorting speed.

In the described methods, a large number of quantitative features are extracted from cell images, to generate a fingerprint of each individual cell. Deep learning, clustering, dimensionality reduction, or support vector machines are then used to classify cells based on these fingerprints. The underlying idea is to correlate geometric parameters such as the size, shape, or distance of the nucleus from the cell membrane, with functional parameters of the cells[8]. However, current high-throughput methods including morphological cell parameter measurements have to compromise between resolution and speed, currently prioritizing either speed[8] or image quality[9,12,13].

Alternatively, scattered light signals provide access to particle or cellular properties without the need for reagents. Light scattered by cells contains a wealth of information that correlates not only with their geometry but also with their cellular structure[14,15], such as mitochondria, peroxisomes, lysosomes, granules, and thus the functional properties of cells[16]. In standard flow cytometry, forward and side scatter measurements (FSC and SSC) are used to distinguish, for example, lymphocytes from monocytes or apoptotic cells. Measuring the scattered light pattern in more detail even allowed the distinction of T and B lymphocytes, monocytes, and cells of different disease stages[12]. However, camera-based light scatter detection is subject to the same limitations of throughput and dynamic range as imaging methods.

We have recently described a flow cytometry setup that has a reduced laser spot size to improve the spatial resolution and detects the forward

[1]German Rheumatology Research Center (DRFZ) - Flow Cytometry Core Facility, Charitéplatz 1 (Virchowweg 12), 10117 Berlin, Germany. [2]APE Angewandte Physik und Elektronik GmbH, Plauener Straße 163-165 / Haus N, 13053 Berlin, Germany. [3]Department of Cytometry, Institute for Biotechnology, Technische Universität Berlin, Berlin, Germany. ✉e-mail: kaiser@drfz.de

scattered light at a wide range of angles using an array of optical fibers connected to photomultiplier tubes (PMTs)[17], which we call multi-angle pulse shape (MAPS) flow cytometry. We use custom-made high frequency signal processing electronics to acquire and store the whole signal waveform of the cell or particle as it transits through the laser beam, the so-called pulse shape. From these complete highly resolved pulse shapes, we extract cell-specific morphological features not accessible in conventional flow cytometers[17–22]. In previous experiments, we have demonstrated the use of MAPS flow cytometry for the label-free identification of the cycle phases of individual cells[17].

Here, we present an extension of this technique that allows sorting of cells and particles based on pulse shape characteristics. As proof of principle, we demonstrate the sorting of cells in different cell cycle phases based solely on their light scattering properties. In addition, we have identified subgroups of T and B cells solely on the basis of the properties of their scattered light.

This cell analysis and sorting method will help to understand how functional cell properties are encoded in the light scattering properties. As such, it may contribute to a better insight into, for example, the state of cells in health or in their pathogenesis on the basis of a label-free measurement.

## Methods
### Sample preparation
**Cell culture**. HEK293 cells were grown as a monolayer in DMEM (Gibco) and Jurkat cells were grown in suspension in RPMI 1640 medium (Gibco), supplemented with 10% fetal bovine serum (FBS, Corning) at 37 °C and 5% $CO_2$.

**Bromodeoxyuridine incorporation**. To detect actively replicating cells, exponentially growing cells were treated for 1 h in fresh medium containing 60 μM Bromodeoxyuridine (BrdU, Biolegend), washed twice with 1x PBS (Th. Geyer) and harvested for fixation. Adherent cells (HEK293) were detached by trypsinization.

**Cell fixation and staining for cell cycle analysis and sorting**. For fixation, 4.5 mL of cold ethanol (70%, −20 °C, Merck) were added to $1 \times 10^7$ cells in 0.5 mL PBS in a dropwise manner while vortexing and incubated overnight at −20 °C. After fixation, cells were centrifuged and washed once in PBS. Cells were then treated first with 2 M HCl/0.5% Triton X-100 (Merck, Sigma) for 30 min at room temperature (RT). After centrifugation, cells were neutralized by re-suspending in $Na_2B_4O_7$ (Merck) and incubation for 2 min at RT. Cells were washed once in PBS/1% BSA and re-suspended in PBS/1% BSA/0.5% TWEEN20 (Sigma) and incubated with anti-BrdU-FITC antibody (Invitrogen) for 1 h at RT. For DNA staining, cells were incubated with propidium iodide (PI) staining solution (PBS containing 50 μg/mL PI (Sigma), 100 μg/mL RNase A (Invitrogen)) for 20 min at RT.

**Hoechst 33342 staining for cell cycle analysis and sorting of live cells**. For MAPS sorting, Jurkat cells were seeded at a concentration of $2 \times 10^5$ cells/mL and ten aliquots of 1 mL were taken from the cell culture. The aliquots were stained successively with Hoechst 33342 (Sigma) at a final concentration of 10 μg/mL for 30 min at 37 °C directly before the analysis or sort. For conventional sorting, Jurkat cells were seeded at a concentration of $1 \times 10^5$ cells/mL and cultivated overnight. Hoechst 33342 was added to the cell suspension to give a final concentration of 5 μg/mL and were incubated for 60 min under cultivation conditions. Cells were harvested by spinning down and washing once with PBS. Propidium iodide was added to the samples to identify dead cells.

**PBMC samples**. Blood samples were obtained from healthy donors in form of leukocyte filters from the Zentrum für Transfusionsmedizin und Zelltherapie Berlin gGmbH (ZTB). Selection of the filters was conducted by ZTB. All ethical regulations relevant to human research participants were followed. The study was approved by the Ethikkommission am

Campus Virchow-Klinikum of the Charité Universitätmedizin Berlin (EA2/084/19). PBMCs were isolated following standard protocol by density gradient centrifugation on Percoll and frozen in freezing medium (DMSO/FCS) at lower temperature than −150 °C. After thawing and washing in warm RPMI (Gibco) medium the PBMCs were counted.

Three different cocktails containing pretitered dilutions of fluorescently-labeled antibodies were prepared: Sample A (CD19 staining) contains antibodies against CD3 (anti human-CD3 FITC, Cat. No. 317306, Biolegend), CD14 (anti human-CD14 FITC, Cat. No. 325604, Biolegend) and CD19 (anti human-CD19 PE, Cat. No. 363004, Biolegend). Sample B (CD4/CD8 staining) contains antibodies against CD3 (anti human-CD3 FITC, Cat. No. 317306, Biolegend), CD4 (anti human-CD4 PE-Cy7, Cat. No. 51-9013251, BD) and CD8 (anti human-CD8 PE, in house conjugated, DRFZ). Sample C (CD45RA/CD45RO staining) contains antibodies against CD3 (anti human-CD3 FITC, Cat. No. 317306, Biolegend), CD45RA (anti human-CD45RA PE-Cy7, Cat. No. 304126, Biolegend) and CD45RO (anti human-CD45RO PerCP Cy5.5, Cat. No. 304222, Biolegend).

Each cocktail of antibodies was mixed with $1.5 \times 10^5$ PBMCs and incubated at room temperature for 20 min. The cells were subsequently washed in PBS containing 2% BSA and resuspend in PBS.

### Conventional flow cytometry and sorting
Conventional cell sorting was performed using a BD FACSAria II cell sorter (BD, San Jose, USA) equipped with 405, 488, 561, and 632 nm lasers. Cell analysis was performed using a BD FACSCanto II (BD, San Jose, USA) equipped with 405, 488, and 632 nm lasers. The Hoechst 33342 staining was detected on the Aria II using a 450/50 nm band pass filter and on the FACSCanto II with a 510/50 nm band pass filter. A 405 nm laser was used for the excitation of Hoechst 33342 in both instruments. On the FACSCanto II, PI and FITC were excited with a 488 nm laser and detected with a 670 long-pass filter and a 530/30 nm filter, respectively. All data were exported in FCS 3.0 format.

### MAPS flow cytometry and sorting
**Optical setup and fluidics**. A custom-built flow cytometry setup (MAPS) was used for cell analysis and sorting. The basis of the setup is an LSR II (BD, San Jose, USA) extended with a BD FACSort (BD, San Jose, USA) "catcher tube" flow cell for sorting[23]. The optical components and the signal processing electronics were replaced by alternative components. Merely the flow cell assembly, the fluidics system, and the optics for side scatter (SSC) and fluorescence light were kept from the original setup. The optical system is outlined in Fig. 1 and descriptions of all optical components are given in Table 1. A 488-nm continuous wave Sapphire (Coherent, Santa Clara, USA) laser with an optical output power of 20 mW is used as the light source. Beam shaping optics with two cylindrical lenses CL1 and CL2, a spherical lens SL1, and two clean-up slit apertures A1 (along $x$) and A2 (along $y$)[24] produce an elliptically shaped

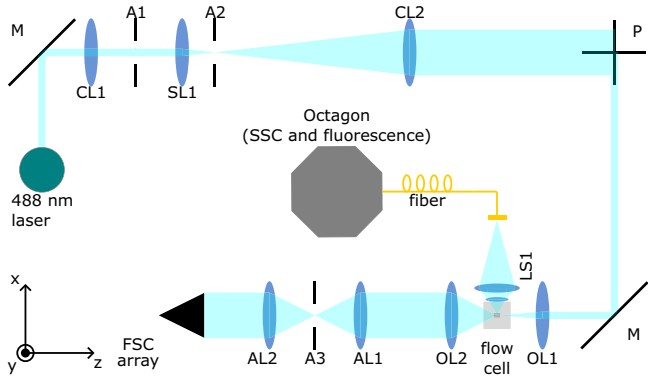

**Fig. 1 | Overview of the optical components of the modified flow cytometry setup.** Descriptions of the optical elements can be found in Table 1.

**Table 1 | Description of the optical components in Fig. 1**

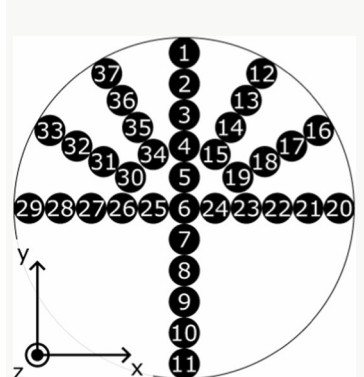

| M, P | Metallic mirrors / periscope | CL2 | Cylindrical lens, $f = 300$ mm |
|---|---|---|---|
| CL1 | Cylindrical lens, $f = 50$ mm | OL1 & 2 | Objective lenses, $f = 20$ mm |
| A1 | 100 µm slit along $x$ | LS1 | LSR II fluorescence/SSC optics |
| SL1 | Spherical lens, $f = 50$ mm | AL1 & 2 | Achromatic spherical lenses, $f = 50$ mm |
| A2 | 100 µm slit along $y$ | A3 | 50 µm slit along $x$ |

The inset on the left shows the layout of the fiber array for FSC light collection.

beam. The long axis of the beam is parallel to the $x$-axis. After passing the periscope P, the cross section is rotated such that the long axis is parallel to $y$, the polarization direction at the interrogation point is along $x$. The sample flow direction is along $y$. For focusing the beam into the flow cell, an M Plan Apo 10× (Mitutoyo, Japan) objective lens OL1 is used. From beam profile measurements of the elliptical cross section before the focusing objective, the spot size behind the objective can be estimated[25]. The estimated $1/e^2$ full width in the focal point amounts to 3 µm along the flow direction of the particles. Fluorescence and side scattered light are collected with the original optics of the LSR II: a gel-coupled lens system LS1, a fiber, and the "Octagon" with optical filters (bandpasses 530/30 and 676/29 nm for FITC and PI detection). The "Octagon" is a detector/filter arrangement with up to eight PMTs coupled to a fiber. It allows to spectrally distribute the incoming light onto multiple detectors. The forward scattered (FSC) light is collected with objective OL2 which is equivalent to OL1. The FSC light is then focused by an achromatic lens AL1 through another slit aperture A3 that is oriented along $x$ direction and collimated again by AL2. Finally, the FSC light is sent to a fiber array with the layout shown in the inset in Table 1. The fibers are connected to photomultiplier tubes (PMTs) R3896 (Hamamatsu Photonics, Hamamatsu, Japan) with 488/10 nm bandpass filters and neutral density filters as necessary. Three fibers at a time were used in the measurements. The respective fiber numbers are given with the results.

The channel cross section in the flow cuvette is rectangular with dimensions $480 \times 180$ µm. The sheath pressure was set to 4.5 psi, which results in a flow speed of the particles in the core stream of around $(5.2 \pm 0.9)$ m/s. The flow speed was determined by optimizing the laser delay settings at different sheath pressures on a Fortessa (BD, San Jose, USA) with a fluidics system of equivalent construction.

**Data acquisition and sort mechanism.** Signal processing and data acquisition was performed by a custom-built system (APE, Berlin, Germany). This system comprises analog amplifiers with 1 MHz bandwidth and power supplies for PMTs connected to a hub that controls the PMT voltages and generates a trigger signal. Digitization and acquisition is performed on a field-programmable gate array (FPGA) board. The signal processing electronics allow for acquisition of whole signal intensity pulse shapes during particle transit through the laser beam in 8 channels. Every channel collects the signal waveforms of a respective scattered light angle or fluorescence detector. The sampling rate was set to 10 MHz with a trigger window length of 8 µs (80 data points per pulse). The acquisition system also calculates wavelet transform coefficients for sort decisions in real-time. The discrete wavelet transform (DWT) is based on the algorithm of the PyWavelets package[26] with adaptions to process a continuous data stream. The wavelet coefficients at scale 8 after transformation with the Haar wavelet are calculated. This results in 5 coefficients per channel and event. The acquired data are delivered in

three files: a binary file with the raw signal intensity data (pulse shapes), a binary files with the wavelet transform coefficients, and an FCS 3.1 file with typical A, H, and W parameters. The sorting is based on assigning each event to one out of up to 8 clusters in each channel. To that end, the coordinates of cluster centroids in wavelet coefficient space are defined and the algorithm decides for each event which centroid is closest (Euclidean distance). A cluster contains events with similar wavelet coefficients and the clustering procedure is explained in the following section. Clusters can be combined from all channels to refine the sort criteria. The calculation of the wavelet coefficients is performed continuously on the incoming data stream with an adjustable time offset. The actual sort decision takes 32 ns. Thus, the time delay between a trigger event and the sort decision is typically below 10 µs and therefore much shorter than the travel time of several 100 µs from the laser interrogation point to the sort point.

The flow cell assembly with the "catcher tube" has two outlets (waste and sort). The mechanical sort unit is located directly above the measurement cuvette, thus the flow direction is upwards as depicted in Fig. 2. When an event fulfills the sorting condition, the FPGA board sends a signal ramp to a piezo driver unit (APE, Berlin, Germany) to move the "catcher tube" into the sort stream and deflect the object of interest into the sorting outlet. The delay between the acquisition trigger and the onset of the sort signal was set to 200 µs and the duration of the signal was set to 400 µs. These values were determined by evaluating the efficiency of a polymer particle sort. The duration of 400 µs includes a signal ramp that spans ≈300 µs and the plateau is followed by a falling ramp of the same duration. The signal processing could theoretically render 50,000 sort decisions a second. However, the sort speed is mechanically limited to around 300 events/s.

**Data analysis, sort preparation, and cluster-sort.** The workflow for sorting is summarized in Fig. 2. For fixed cells, a measurement of $5 \times 10^4$ events was performed with the MAPS setup on the sample intended for subsequent cluster sorting. Gates for the cell cycle phases (G1, S, G2/M) were defined on the staining intensity (PI, BrdU-FITC) with FCSExpress 7 (De Novo Software, Pasadena, USA) and their event indices were exported to a text file. For live cells, a Hoechst 33342 staining was used which cannot be detected with the MAPS setup. Therefore, the sample was split into a cluster definition sample and a sort sample. The cluster definition sample was sorted for G1, S, and G2/M phases by conventional cell sorting with a FACSAria II based on the Hoechst 33342 staining. The fluorescence-sorted cells were analyzed with the MAPS setup and a merged data file was created from the three individual datasets (G1, S, G2/M). Around $1–3 \cdot 10^4$ events were acquired per cell cycle phase. The merged file contains a file ID for each event that provides the information on the cell cycle phase based on staining. Next, dead cells and debris were excluded by gating on scattered light signals

https://doi.org/10.1038/s42003-024-06759-5                                                                **Article**

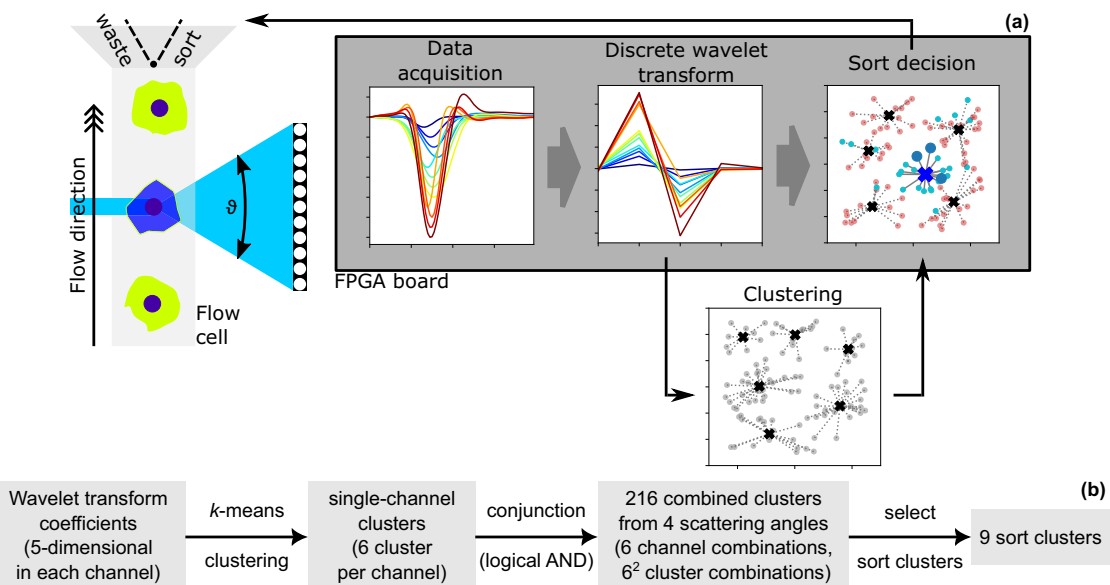

**Fig. 2 | Custom built MAPS (multi-angle pulse shape detection) flow cytometry setup with cluster-based sorting capabilities. a** From each object transiting through the laser beam, fluorescence and scattered light pulse shapes in multiple scattering angles are acquired. An on-board wavelet transform is performed. Data is stored in binary files. Following offline analysis, cluster centroids are fed back to the signal processing electronics and can be used for sorting. **b** The process of clustering, combining clusters from different channels, and selecting clusters for sorting.

(FSC fiber 23 and SSC, similar to a conventional flow cytometer). As for fixed cells, events were gated for the cell cycle phases (here indirectly using the file ID) and these indices were exported to text files. Within a custom Python[27–30] script, raw pulse shape data, wavelet transform coefficients, the FCS file, and the text files with the gate indices (from staining or file ID) are loaded. Then, a $k$-means clustering[31] was performed on the wavelet transform coefficients of the scattered light channels as given in Fig. 2b. All events were included in the clustering. Subsequently, all possible conjunctions (logical AND) of single-channel clusters from two different channels were generated. These combined clusters were analyzed in terms of how the events distribute into the gates defined before with the aid of the PI/BrdU-FITC staining (fixed cells) or the file ID (live cells). A ranking was generated which displays the three combined clusters with the highest enrichment for each of the cell cycle phases. Some of the highly enriched clusters consist of very few events and are thus unsuitable for sorting due to dilution and long sorting times. Thus, a second ranking was generated with the largest clusters above a defined enrichment threshold. This threshold was set to 75% (fixed cells) or 67% (live cells) of the maximum achievable enrichment. From these two rankings, three sort clusters for each phase were selected. The cluster centroid coordinates were transferred to the signal processing electronics. In the case of fixed cells, a single sample was used for all sorts of the MAPS experiment. The sample was vortexed after each individual sort. In the case of live cells, the sorting sample was split again and Hoechst-stained immediately before the MAPS sort to reduce cell death. For each sort cluster, 45–50 mL of sorted sample were collected. The sample collection tubes were coated with PBS/BSA (2% w/v) before sort. Due to the sort mechanism, there is a strong dilution resulting in low cell concentration. At the beginning of each cluster sort, the sort outlet was redirected into the waste for 30 s to fill the tubing with the desired cells and avoid cross-contamination.

**Re-analysis of sorted cells**. The sorted samples were centrifuged and re-suspended in a smaller volume. Subsequently, samples were analyzed with a FACSCanto II for sort purity. The success of the MAPS-sorting of fixed cells was validated using BrdU-FITC and re-staining with PI to

determine the fractions of cells in the cell cycle phases. Live cells were stained with Hoechst 33342 before the MAPS sort. We define the enrichment $x(P, c)$ of a cluster $c$ with cells from cell cycle phase $P$ as

$$x(P, c) = \frac{\left(\frac{N(P, c)}{N(\cdot, c)}\right)}{\left(\frac{N(P, \cdot)}{N(\cdot, \cdot)}\right)} \tag{1}$$

where $N(P, c)$ is the absolute number of cells from cluster $c$ that fall into the gate of cell cycle phase $P$. The cell cycle phase $P$ is one of the phases G1, S, and G2/M. The cluster $c$ denotes the sort clusters. The symbol $\cdot$ is a placeholder for all phases or clusters (e.g., $N(\cdot, \cdot)$ means all single cells of the whole sample). Thus, for each cluster there are three values of $x$, one for each cell cycle phase.

## Statistics and reproducibility

The reproducibility of cell sorting with the described setup is qualitatively given by multiple repetitions of the cell sorting procedure with different samples from two different cell lines over the course of 6 months. Moreover, comparable results were obtained with an earlier version of the setup[17]. Compared to conventional flow cytometers, the high-precision optics are more prone to temperature drift. Therefore, a quantitative long-term assessment using the same clustering data was not aimed at, yet.

## Results

As described in Methods Section, fixed HEK293 cells and live Jurkat cells were sorted with wavelet coefficient clustering as a proof of principle for MAPS sorting. Fixed cells were stained with PI and BrdU-FITC, analyzed with MAPS for sort cluster definition, sorted, and re-analyzed. Live cells were stained with Hoechst 33342, and sorted with a conventional sorter. The sorted samples were used to define the MAPS sort clusters, cells were sorted and re-analyzed. The fluorescent staining was only used for cluster definition and re-analysis, not for the MAPS sorting itself.

### Fixed cell sort

Cell sorts of HEK293 cells on time-resolved FSC and SSC light scatter signals at different scattering angles were performed. The results of a representative experiment from multiple replicates are presented here. The FSC fiber

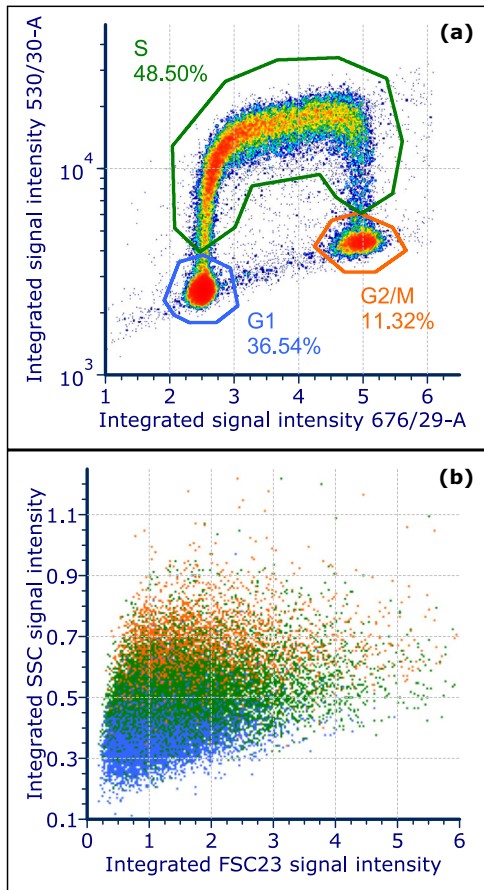

**Fig. 3 | Characterization of cell cycle phases of HEK293 cells with the MAPS setup. a** Number density plot for intensity distribution of BrdU-FITC against PI. Gates were set for the cell cycle phases G1, S, and G2/M. The events displayed were gated for single cells based on pulse width and area of the PI fluorescence signal. **b** Overlay dotplot for scattered light intensity of the cells within the gates defined in (**a**).

numbers used were 23, 8, and 4 (cf. Table 1). The fluorescence intensity distribution of single cells (cf. Fig. S1) stained by BrdU and PI is displayed in Fig. 3a. As expected, three populations can be clearly distinguished. The gates to define the G1, S and G2/M phases are shown with their respective fractions in the single cells. Figure 3b shows the scattered light (FSC fiber 23, SSC) intensity distribution comparable to conventional FSC-SSC plots. Due to the overlap of the cell cycle subsets, conventional scatter plots are not sufficient to distinguish the cell cycle stages.

Figure 4a–d displays exemplary pulse shapes (semi-transparent lines) of individual cells within the gates defining G1, S and G2/M phase, respectively, for the SSC channel and the three FSC angles together with the averaged pulse shapes (solid lines). The pulses show a variability in the pulse shapes between individual events and gates. However, on average, the pulse length increases from G1 over S to G2/M in all channels. In SSC (Fig. 4a) and FSC23 (Fig. 4b), also the pulse height varies specifically while in FSC8 (Fig. 4c) a slight change in pulse shape at the onset of the pulse and a clear shift of the pulse peak were observed. In FSC4 (Fig. 4d), the strength of absorption was slightly lower for later cell cycle phases.

The discrete wavelet transform (DWT) coefficients averaged within each gate are shown in Fig. 4e–h. In SSC (Fig. 4e), the G2/M phase cells were clearly separated from the other two cell cycle phases due to lower coefficient magnitudes. In FSC23 (Fig. 4f), also the course of the coefficients differs between cell cycle phases. For FSC8 (Fig. 4g), the wavelet coefficients were rather similar between the phases. Finally, in FSC4 (Fig. 4h), the G1 phase cells could be clearly separated from S and G2/M.

In order to find specific pulse shapes and scattering angles important for distinguishing cell cycle phases, we applied a *k*-means clustering algorithm to the wavelet coefficients. The workflow is illustrated in Fig. 2b. In each scattering angle, six clusters were defined to group similar pulse shapes. The six single-channel clusters from different angles were combined in conjunction (logical AND). Thereby, we obtained combined clusters from the four scattered light channels (SSC, FSC23, FSC8, FSC4). The distribution of the clustered events into the cell cycle phase gates, cf. Fig. 3a, was analyzed. Nine combined clusters with high enrichment with cells from a certain cell cycle phase were selected for sorting. An overview of these sort-clusters is given in Table 2. The listing shows which light scatter channels contributed to the sort-clusters. For example, the sort-clusters that were specific for G1 seem to be dependent on features in the SSC signal since they all contain a single-channel cluster from SSC. The S phase sort-clusters were mainly defined by single-channel cluster 3 in FSC4. For G2/M, SSC single-channel cluster 2 and FSC8 cluster 5 were important.

The sort clusters defined above were used for a label-free cell cycle sort. The sort mechanism was triggered when an event fell into the sort cluster selected from Table 2 (cf. MAPS flow cytometry and sorting Section). To illustrate this procedure, Fig. 5 shows the six single-channel cluster centroids and 100 exemplary events in a 2D projection from 5D wavelet coefficient space for two scattering angles. Figure 5a displays the wavelet coefficients 1 and 2 from SSC pulses, Fig. 5b displays the wavelet coefficients 2 and 3 from FSC4 pulses (see Fig. 4e, h) for the full 5D wavelet coefficient averages). The lines show to which cluster an event was assigned. The two single-channel clusters marked in blue are those that constructed the sort cluster. This example uses sort cluster 6 for G1 which contained of the single-channel clusters 0 and 4 from SSC and FSC4, respectively (see Table 2). Events that matched the light scattering properties (displayed with a blue and larger dot) pre-defined by the sort cluster were sorted. Events that were only assigned to one of the single-channel clusters of the sort cluster are displayed in gray and are termed "partially matching," the red dots are events that were assigned to single-channel clusters that were not part of the sort cluster in both channels. In both cases, they were not sorted.

For sorting, the cluster centroid coordinates were transferred to the signal processing electronics. Cells from each of the nine sort clusters (Table 2) were sorted with the MAPS sorter. The sorted samples were re-analyzed with a FACSCanto II flow cytometer and compared to an unsorted aliquot (see Fig. S2). Sorting of G1-specific sort clusters (Fig. 6a–c) resulted in a twofold enrichment of cells in G1 compared to the unsorted sample. Cells in G2/M phase were almost completely depleted, while cells in S phase were clearly reduced but not completely expelled from these clusters. The S phase cells included in the G1 cluster were still in early S phase with integration of BrdU but minimal increase in PI staining. In the sorted G2/M clusters (Fig. 6g–i), cells in G2/M phase were enriched by a factor of 4.5–5. Cells in G1 phase were efficiently depleted. Cells in S phase were reduced by 25% to 50%. The remaining S phase cells were in late S phase with increased BrdU integration and high PI signal. The sort clusters defined for cells in S phase were less specific (Fig. 6d–f). In all three S-phase sort clusters, considerable fractions of G1 and G2/M cells remained, with an enrichment factor of cells in S phase of only around 1.4. While the fractions of cells in G1 phase were generally reduced in the S phase clusters, there was even a co-enrichment with G2/M cells (Fig. 6e, f) compared to the unsorted sample.

## Live cell sorts

Jurkat cells were used for live cell sort experiments. The cluster definition and MAPS sort were performed as described in MAPS flow cytometry and sorting Section and in analogy to Fixed cell sort Section. As before, nine sort clusters were selected for the sort, three for each cell cycle phase. The fraction of cells covered by a sort cluster cannot be determined directly since the cluster analysis is applied to an artificially merged dataset. In this merged data set, the fractions of cells from the three phases do not represent the original sample. However, an extrapolation shows that each sort cluster

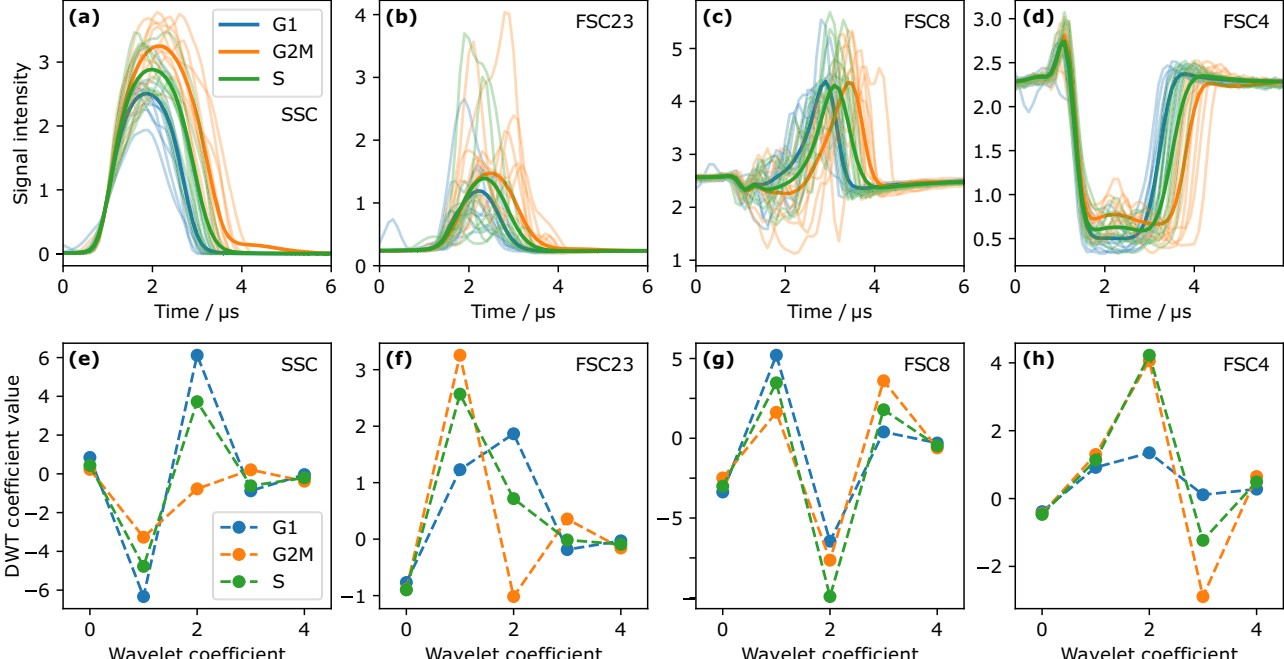

**Fig. 4 | Pulse shape and wavelet coefficient analysis of HEK293 cells.** Averaged pulse shapes (**a–d**) and wavelet coefficients (**e–h**) from the events within the gates defined in Fig. 3a. The semitransparent lines in (a-d) show exemplary pulses of individual cells, the solid curves are averages of all pulse shapes of cells within the indicated cell cycle phase. The numbering of the FSC channels refers to the layout in

Table 1. The average pulses mainly differ in pulse height and length but depending on the channel also in the actual shape. The wavelet coefficients exhibit clear differences between the cell cycle phases. However, depending on the respective channel, often two neighboring phases show more similarity.

covers roughly 5–8% (G2/M) or 12–17% (G1, S) of all cells. Please note that the sort clusters are not mutually exclusive. MAPS-sorted samples were re-analyzed with a FACSCanto II. The resulting distributions in the cluster-sorted samples representing G1, S, and G2/M based on DNA quantification with Hoechst 33342 are displayed in Fig. 7a–c. The G2/M cluster-sorted samples reached purities of 73-85%, those for G1 had purities of 93-96%, corresponding to an enrichment by a factor of around 2.5 for G2/M and around 2 for G1. For comparison of MAPS with conventional cell sorting on scattered light, Jurkat cells were also sorted using gating for FSC-SSC intensities with a FACSAria II cell sorter. To this end, back-gating from the Hoechst intensity was used to determine their FSC-SSC intensity. From this, FSC-SSC gates were defined for sorts. The gating strategy is displayed in Fig. S3. The results of the re-analysis is displayed in Fig. 7d–f. The sort for G1, Fig. 7a, reveals specificity for G1 cells. However, with around 90% it is less pure than the MAPS-sorted samples. For the S phase (Fig. 7e), the sort failed since the resulting Hoechst intensity distributions did not show any specificity. In the case of G2/M (Fig. 7f), there was a clear tendency toward G2/M cells but the sorted sample still contained significant fractions of S and also G1 cells and only reached a purity of under 65%. In general, the sorting

with MAPS allowed for a more stringent selection of cells in the different cell cycle phases compared to the classic FSC-SSC sorting.

**Analysis of PBMC subsets**

In addition, we have used our technology for a proof-of-concept analysis to identify B cells and subsets of T cells of peripheral blood mononuclear cells (PBMCs). To this end, three samples were prepared: (A) CD3-FITC, CD19-PE, CD14-FITC, (B) CD3-FITC, CD4-PECy7, CD8-PE, (C) CD3-FITC, CD45RA-PECy7, CD45RO-PerCPCy5.5. For the measurement of the three samples, the optical setup from MAPS flow cytometry and sorting Section was slightly modified by a further fiber array for the angle-resolved detection of the SSC light. The signal processing was not changed. As before, *k*-means clustering was used to find similar pulses derived from multiple angles of FSC and SSC. However, since no sorting was sought, the raw pulses of scattered light channels were clustered. The obtained clusters were screened for specificity for the subsets stained by fluorescent antibodies in the respective sample. For sample A (CD19 staining), events were gated for lymphocytes (in SSCI-FSC23-A) before clustering. Clusters with enrichment and depletion of B cells (CD19-positive) were found. The distribution of the staining intensity of the lymphocytes resulting from the two exemplary clusters are shown in Fig. 8a–c. For sample B (CD4/CD8 staining), clustering was performed on CD3-positive events. Two example clusters were selected that show an enrichment for either CD4 or CD8 cells as displayed in Fig. 8d–f. For sample C (CD45RA/RO staining), again only CD3-positive events were selected for clustering. Therein, combined clusters were identified that were enriched for either CD45RA or RO cells. This is given in Fig. 8g–i.

**Discussion**

We present a custom-built flow-cytometric cell sorter that captures the full signal pulse shapes during cell transit at multiple scattering angles. The sorting uses wavelet transform coefficients that are calculated from the signal pulse shapes. Each event is assigned to one out of six user-defined cluster centroids separately in each measurement channel and the clusters

**Table 2 | Overview of the nine sort clusters**

| Sort Cluster | 1 | 2 | 3 | 4 | 5 | 6 | 7 | 8 | 9 |
|---|---|---|---|---|---|---|---|---|---|
| SSC cluster | --- | 2 | 2 | 0 | 3 | 0 | 0 | --- | --- |
| FSC23 cluster | --- | --- | --- | --- | --- | --- | --- | 0 | 4 |
| FSC8 cluster | 5 | 5 | --- | 0 | 0 | --- | --- | --- | --- |
| FSC4 cluster | 3 | --- | 0 | --- | --- | 4 | 3 | 3 | 3 |
| % of single cells | 1.71 | 3.64 | 3.82 | 3.73 | 11.51 | 4.08 | 12.8 | 5.53 | 3.01 |
| Phase | G2/M | G2/M | G2/M | G1 | G1 | G1 | S | S | S |

The listing shows the numbers of the single-channel clusters that were used to form the sort clusters. The size of the clusters is given as the fraction of single cells.

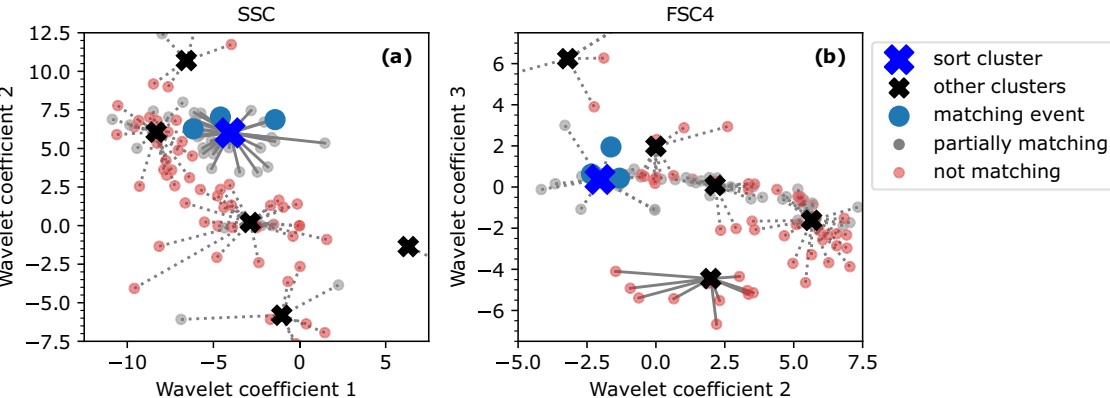

**Fig. 5 | Projection of wavelet coefficient space showing the sort decision process.** Illustration of the sort decision process in a 2D projection of the 5D wavelet coefficient space for two channels: (**a**) SSC, (**b**) FSC4. The six cluster centroids are shown along with 100 exemplary data points from measured cells. The highlighted sort cluster is one that was used for G1 sorting. The red dots represent events that did not match the cluster selection in both channels. The gray dots represent events that matched only one single-channel cluster. The larger blue dots are those events that would have been sorted.

**Fig. 6 | Fluorescence intensity distributions of samples sorted for cell cycle phases (BrdU-FITC/PI).** Upper row (**a–c**): clusters selected for G1 specificity. Middle row (**d–f**): clusters selected for S specificity. Lower row (**g–i**): clusters selected for G2/M specificity. The displayed events are gated for single cells. The percentages indicate the fractions of the events in the gates with respect to the number of single cells. The factors denote the enrichment of the sort cluster with cells from the respective cell cycle phase as defined in Eq. (1). Panel titles denote cluster channels and numbers.

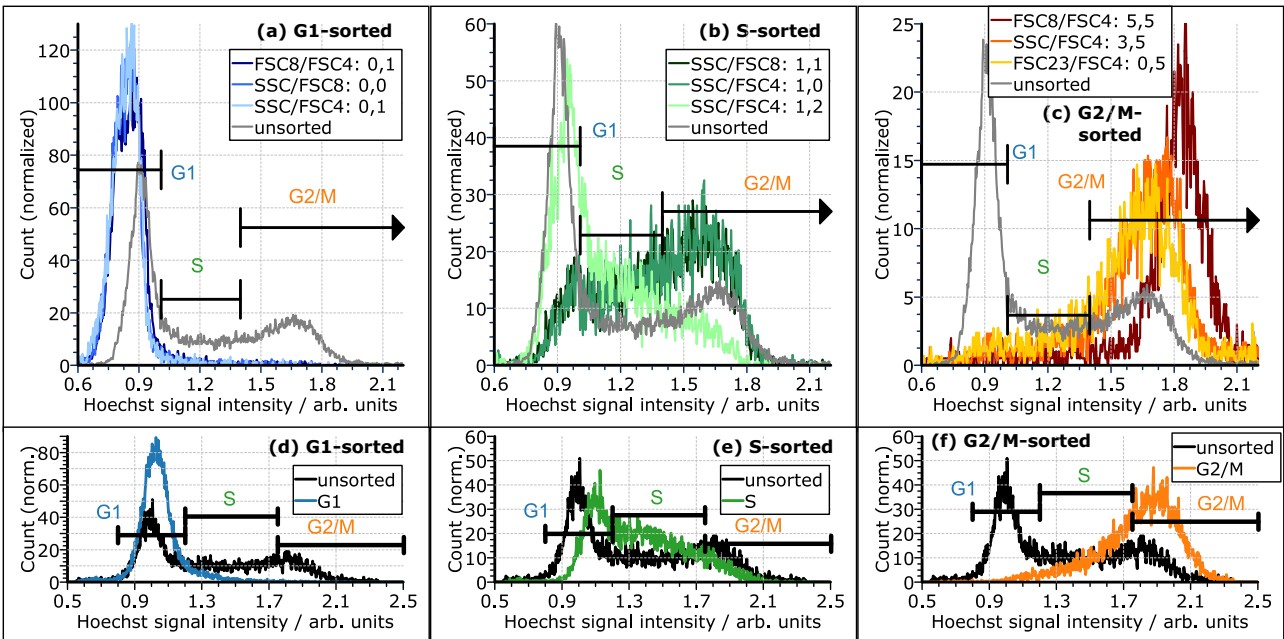

**Fig. 7 | Hoechst 33342 intensity histograms of samples sorted with MAPS and on FSC-SSC with a conventional cell sorter. a–c** Samples that were MAPS-sorted for G1, S, and G2/M based on sort clusters. The intensity distribution of the unsorted sample is shown for comparison. Three clusters were sorted for each cell cycle phase. **d–f** For comparison, a sample was sorted by gating in standard FSC-SSC intensities with a conventional sorter.

can be combined to specify the sorting criteria. This definition of sorting criteria differs from the standard hierarchical gating approach in that it uses boundaries between clusters in a multi-dimensional space of wavelet coefficients. We suppose that this approach is more suitable to represent continuous biological processes. MAPS also enables label-free sorting and as a proof of principle, we demonstrated the sorting of cells according to their cell cycle phase. Sort performance was verified with established staining methods.

Based on our previous work on label-free cell cycle analysis with scattered light[17], we now introduce sorting capabilities using wavelet transforms. This includes the modification of the clustering procedure to define sort clusters. Sorting fixed HEK293 cells revealed specificity of the sort clusters for G1 and G2/M phase cells. The G1-sorted samples still contained some early S phase cells and the G2/M sort clusters had residual late S phase cells. In both cases, these cells could be assigned to the S phase due to increased BrdU incorporation but hardly with the PI staining alone. In the G2/M sort clusters, there may have been cells present that incorporated BrdU but then transitioned to G2/M during the 1 h incubation time. Such cells may have been correctly identified by the MAPS sorter as G2/M but were falsely classified as late S phase cells by the staining. The three S-sorted samples were enriched for the S phase, but two out of three also showed slight enrichment for G2/M.

Interestingly, the S-phase sorting clusters were all based on pulse shapes from the FSC4 channel and two of the three G2/M sort clusters also relied on this channel. As can be seen in Fig. 4d, the signal in FSC4 is governed by extinction due to the cell absorbing or deflecting light away from the optical axis. Thus, the light extinction seems to carry characteristic features of the S and G2/M phase. Conversely, the SSC played a major role for the characterization of the G1 phase, but is less important especially for the S phase and also for G2/M. This is somewhat surprising since the SSC signal intensity of the G2/M and S phase of a conventional cytometer was slightly higher than from the G1 phase. The increase in SSC intensity was probably a consequence of the increase in granularity of cells in the S and G2/M phases. Therefore, it may be useful to replace the integrating SSC with a fiber array as well. The angle-resolved measurement may provide more details on internal cell structure from the SSC signals.

Furthermore, FSC23 is a specific scattered light angle only for the detection of the S phase. FSC23 detects light scattered in the plane perpendicular to the flow direction. This direction is blocked by the blocker bar (beam stop) in most conventional cytometers. The sort clusters for G1 and G2/M phase cells have no common single-channel cluster. Moreover, since they contain single-channel clusters from FSC4 and 8 but not from FSC23, light scattering and extinction along angles in the plane defined by the optical axis and the flow direction are distinctive for G1 and G2/M. The MAPS-sorted subsets differed in their scattered light properties even between clusters sorted for the same cell cycle phase. This may indicate that discrete class labels introduced by the gating on PI and BrdU signal intensities underrepresent the continuous process of cell division[32].

In general, one has to keep in mind that light scatter and Hoechst or PI staining are fundamentally different: the MAPS analysis is based on morphological properties and the refractive index of the cell while these stainings indicates DNA content. The refractive index of the nucleus was shown to specifically vary during cell proliferation and is considered an indicator for cumulative nuclear density changes resulting not only from DNA but from any macromolecule (e.g., RNA, proteins)[33]. Thus, a perfect agreement between the two methods is not to be expected.

Since cell fixation is known to drastically change the optical properties of cells, we also analyzed and sorted live cells to verify applicability. Sorting live cells produced clean G1- and G2/M-sorted samples. In similarity to the fixed cells, the S-sorted samples were of low purity and contained a significant fraction of cells from the neighboring phases. The re-analysis is reliant on Hoechst staining which provides weaker resolution for the S phase. Thus, a possible minor fraction of S-phase cells especially in the G1-sorted samples may not be seen with the Hoechst staining which led to an overestimation of the purity of G1- and G2/M-sorted samples. Additionally, a conventional sort was required to define the MAPS sort clusters. The concrete choice of the sorting gates therefore also influenced the clustering. The BD FACSAria II cell sorter, used to pre-sort live cells for subsequent cluster selection, was only equipped with a 405 nm laser that is sub-optimal for excitation of Hoechst. Consequently, this affects the specificity of the cluster selection for the MAPS sorting. In addition, Hoechst has been shown to induce single-strand breaks in DNA[34], which results in an increasing

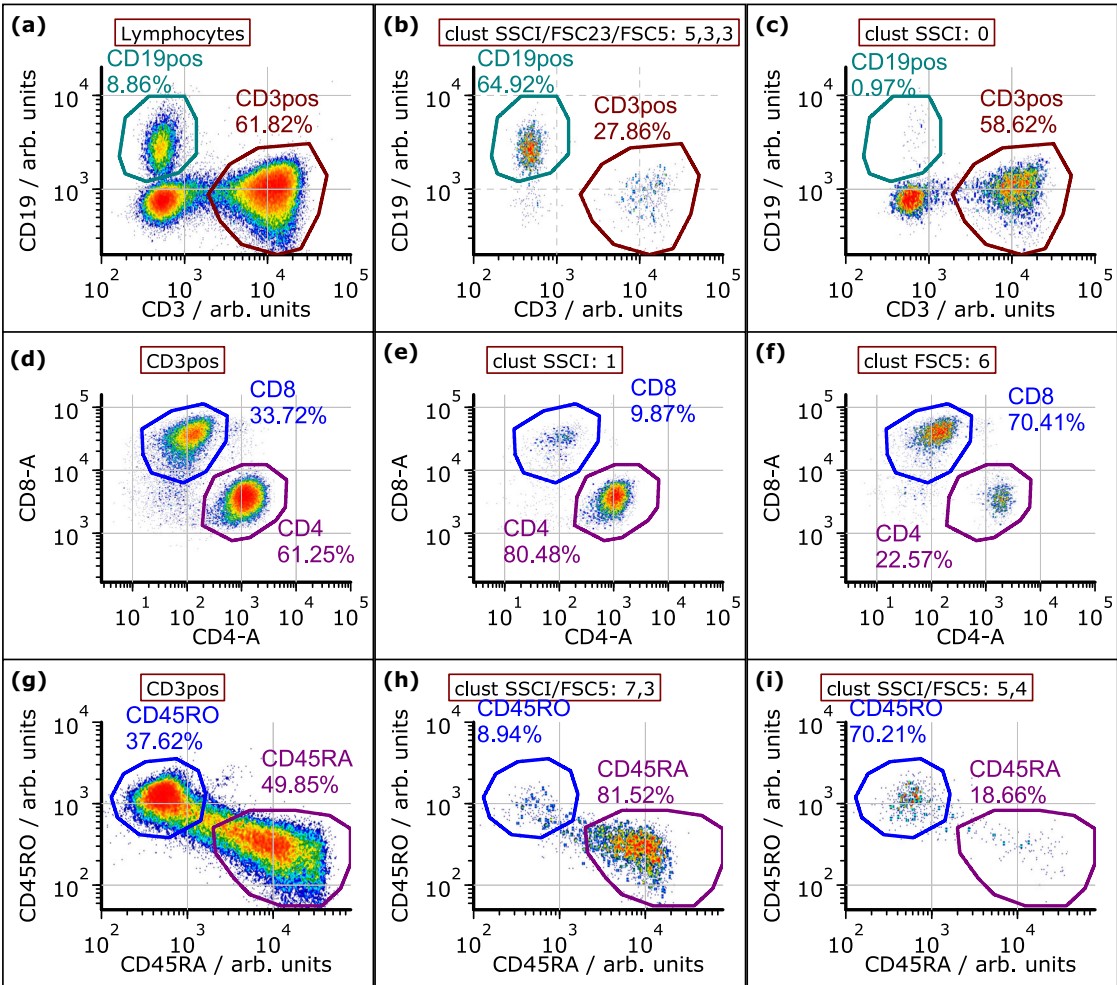

**Fig. 8 | Analysis of PBMC clusters.** A k-means clustering was applied to the full pulse shapes of PBMCs. Exemplary cluster were chosen for enrichment with certain subsets by comparison with conventional staining. **a–c** The clustering was only applied to lymphocytes. Clusters enriched with or depleted of B cells (CD19) were found. **d–f** The clustering was applied to CD3-positive cells. Therein, clusters enriched for either CD4- or CD8-positive cells were found. **g–i** The clustering was applied to CD3-positive cells. Clusters specific for memory and naive cells were found.

number of dead cells over time and thereby also affects the definition of sorting clusters and the composition of the MAPS-sorted samples. Generally, with increasing sort specificity, the recovery decreases.

As a proof-of-concept of further applications, we analyzed PBMC subsets only by means of scattered light and compared the results to conventional immunological staining approaches. Clusters of scattered light pulses with specificity for B cells, CD4, CD8, as well as memory and naive T cells were observed. On the one hand, the analysis conditions were simplified due to a pre-selection of, e.g., only CD3 cells and there is no strict separation of the targeted cell subsets between the clusters. On the other hand, the authors are not aware of another label-free technique with similar capabilities. With further improvements to the data analysis such as artificial intelligence, we expect advances in the label-free identification of PBMC subsets.

The examples presented here are based on established staining methods to demonstrate the specificity of MAPS sorting. However, there are several ways to develop and use MAPS for unsupervised identification of cells or other particles. MAPS provides the prerequisite for the long-term development of databases with pulse shapes. With such databases, cells could be identified only using on intrinsic characteristics, i.e. additional tagging of cells could be omitted. In addition, the MAPS method can be employed for, e.g., label-free investigation of cell function on the basis of morphological cell characteristics and thus for a deeper understanding of cell labeling with biomarkers[35].

For future work, it is planned to systematically evaluate the specificity of the scattered light signals in different scattering angles, e.g. the side scatter angles, for different cell types. We observed a high sensitivity of the obtained pulse shapes to the exact placement of the array, especially in fibers close to the outer region of the transmitted beam. Here, interference of the transmitted beam and the scattered light plays a major role and small deviations in the position can result in large differences in the obtained signals. This is also the reason why the signals in FSC fibers 4 and 8 are not close to symmetrical. Therefore, further experiments are needed to better understand the optimal position of the fiber array with respect to the specificity of the measurement.

The sorting mechanism used only served as a proof-of-principle. Replacing the sort mechanics by a state-of-the-art droplet sorting system is compatible with the data processing in terms of timings and computation speed without the need to adapt sample throughput or flow speed. This would enable a higher sorting speed and higher cell concentration of the sorted cells in the collection tube. Thus, the label-free approach presented here has the potential to overcome throughput limitations of other morphology-based approaches. Due to the use of separate detectors for different scattered light angles, the dynamic range is superior to a single camera. Moreover, the measurement principle is simple and technically compatible to many existing flow cytometers.

Apart from the cell cycle measurement presented here, angle-resolved pulse shape detection used for MAPS in combination with clustering also provides a very simple way to identify cell doublets, dead cells and cell debris.

## Data availability

The datasets generated and analyzed during the current study are available from an open repository (https://doi.org/10.5281/zenodo.10282700)[36]. Any remaining information can be obtained from the corresponding author upon reasonable request.

## Code availability

The computer code used for the presented study is available from an open repository (https://doi.org/10.5281/zenodo.10282700)[36]. The code can be used freely without any restrictions under the MIT License. However, third-party software with restrictive terms of use may be required for execution.

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

## Acknowledgements

All authors thank Stefan Stein and Jochen Barths for providing additional FACSort catcher tubes. This research project is funded by the Deutsche Forschungsgemeinschaft (DFG, German Research Foundation)–491069896, EFRE Project 1.6/01, and the Dr. Rolf M. Schwiete Foundation. D.K. acknowledges support from the Innovators Program of the International Society for Advancement of Cytometry (ISAC).

## Author contributions

D.K., T.K., K.H. and J.K. conducted the flow cytometry measurements. K.H., J.K. and A.W. prepared the cell samples. T.K., D.K. and K.v.V. planned and set up the modified optics. A.E. and J.P. developed the signal acquisition and sorting electronics. D.K. and T.K. performed data analysis. T.K. and H.-D.C. conceptualized the project. All authors were involved in data interpretation and discussion.

## Funding

## Competing interests
A.E., J.P., and K.v.V. are associated with APE who provided the data acquisition electronics and might benefit as a company from an interest in the presented method. All other authors declare no competing interests.
