## [Peer review file · Communications Biology]

Reviewers' comments:

Reviewer #1 (Remarks to the Author):

Kage et al. present an innovative flow cytometer optical configuration enabling the label-free sort of cells based on four scattered light detection dimensions, resulting from multiple angle collection optics. This represents a novel way to define clusters of cells based on the analyses of current pulses analysed using pre-validated pipelines. The authors nicely demonstrate that in the case of a very simple experimental design (i.e., cell cycle phase discrimination), the use of four scattered light detection angles (channels) suffice to identify different cell cycle phases and sort, with some precision, the cells of interest.

Although innovative, it is unclear to this reviewer what is/which are the strategic advantage(s) of the multiple angle pulse shape detection sort. It seems that has strong caveats related to excessive sample dilution (45-50 mL of final sorted cells' volume) for a comparable reduced number of parameters. The strategic advantage seems to be the identification of cells in a label-free condition, which I think is a great step in the right direction, and potentially, a game changer that could enable step-change improvements in the way we analyse and isolate events. Although the data with cell lines is convincing, these rarely represent the true complexity of cell populations needing isolation for downstream functional studies. In other words, this work will benefit greatly by the proof-of-concept sort and subsequent analyses of more complex samples, such as PBMCs (<https://www.biolegend.com/en-gb/products/veri-cells-pbmc-12260?GroupID=BLG2181> or other source of multiple cell types, like mouse spleens). Can this instrument replace or complement conventional sorters by truly identify cells, and accurately sort them, based on their light scattering properties? Can you identify monocytes, B cells and T cells based only on multiple angle scattering properties? Are T cells, for instance, producing clusters with strong association to their phenotypical diversity?

Also, what is the function of the octagon array of PMTs? How many channels have been successfully tested with MAPS sorting? One huge limitation is the processing/computing power needed to resolve the clusters in the 10 microseconds needed between triggering and sorting. What is the data processing limit of the sort with 8 fluorescent channels? Can be circumvented by reducing the flow rate of the sample?

Minor points:

Line 11-12: The known effect of sorting with the use of fluorescence is not backed up by any reference. This reviewer has successfully sorted cells for experiments requiring

subsequent culture, many times. If the authors should provide a reference, data to benchmark survival of cells sorted with their cytometer (compared to a conventional one), or alternatively, remove such statement.

Line 86: the shape of lenses, beams and long axis should be consistent in the diagram.

Line 88: what is the polarisation direction with respect with the sample flow at the interrogation point?

Figure 2: Please correct figure as it seems the direction of the flow and sort is upward.

Line 146: In the data availability statement authors should provide access to the Python script referenced (add link or ID).

The gating strategy for Figure 6 is incomplete. For clarity, the input sample should be shown together with the resulting 9 clusters sorted a-i. Where is the evidence of S, G1 and G2/M being defined over a bi-parametric distribution of singlets?

Reviewer #2 (Remarks to the Author):

In this paper, the authors test the feasibility of their cell sorting technology whose decision is based on the analysis of multi-angular forward scattering using a fiber array and clustering methods employing multi-dimensional space of wavelet coefficients. They demonstrate the sorting of cell lines for their cell cycle stages, and the resultant sorted cells are confirmed by conventional flow cytometry.

I have a feeling that the demonstration of distinguishing cell cycles was not greatly impressive but this technique is neat and it is interesting to see that lots of different information is averaged out in conventional forward scattering signals. With the following comments addressed appropriately, there may be a chance for acceptance in Communication Biology.

1. In line 15, this description is not accurate “The custom-built flow cytometry setup comprises fiber arrays in forward and side scatter detection for angular resolution” because the authors show the multiangular detection in only forward direction. This was confusing and descriptions have to be more accurate in many places.

2. In line 30, I understand that the concept of this technology has similarity or an analogy with ghost cytometry, especially in its label-free versions which integrate

forward scattering waveforms (Ota et al, Science 2018, Ugawa et al, eLife 2021, Tsubouchi et al, Cell Rep Methods 2023) as feature extraction-free and image-quality-free high-content analysis methods. MAPS splits the optical signals in the frequency space, while ghost cytometry does it in the real image space. With references, it is constructive to address ghost cytometry and explain what is advantages and disadvantages.

3. In line 11, this statement can be more accurate: “State-of-the art flow cytometry relies on fluorescent stains to detect the cells of interest. (<https://deepcell.com/>, <https://thinkcyte.com/> and so on)

4. In line 201, it is interesting to see that the responses of channels 4 and 8 are so different. Can authors explain what is the possible reason fundamentally? If possible, can authors take an optical image of cell signals coming at a plane of “FCS away” in Figure 1? Is there no speckle effect? In addition, what is DWS in the legends of y-axis?

5. In line 262, can I see the “gating for FSC-SSC intensities with a FACSAria II cell sorter”? I hope to see what happens if the authors apply a gate which is as good as possible in a reasonable manner.

Response to reviewers

Referee comment	Response by the authors
1. 1. Although innovative, it is unclear to this reviewer what is/which are the strategic advantage(s) of the multiple angle pulse shape detection sort. It seems that has strong caveats related to excessive sample dilution (45-50 mL of final sorted cells' volume) for a comparable reduced number of parameters.	Referee 1 We extended our discussion by a paragraph addressing advantages and disadvantages compared to other label-free cytometry technologies as well as options for upgrading the sorting mechanism. Line 373-378: The sorting mechanism used only served as a proof-of-principle. Replacing the sort mechanics by a state-of-the-art droplet sorting system is compatible with the data processing in terms of timings and computation speed without the need to adapt sample throughput or flow speed. This would enable a higher sorting speed and higher cell concentration of the sorted cells in the collection tube. Thus, the label-free approach presented here has the potential to overcome throughput limitations of other morphology-based approaches. Due to the use of separate detectors for different scattered light angles, the dynamic range is superior to a single camera. Moreover, the measurement principle is simple and technically compatible to many existing flow cytometers. The authors acknowledge that the strong dilution is a critical caveat of the current system. However, state-of-the-art droplet sorting can be used with our data processing without the need of reduced throughput or flow speed. Though the implementation of such a sort mechanism is a sophisticated engineering task, it does not bear fundamental issues. We extended the descriptions in the methods Section and added a paragraph in the Discussion to address these concerns. Line 148-149: The signal processing could theoretically render 50,000 sort decisions a second. However, the sort speed is mechanically limited to around 300 events/s. The number of parameters is not directly comparable to fluorescence-based flow cytometry. The pulse shapes contain information that is more detailed. Adding channels for additional fluorescence measurements is merely a question of upscaling and thereby cost as in any other instrument.
2. Although the data with cell lines is convincing, these rarely represent the true complexity of cell populations needing isolation for downstream functional studies. In other words, this work will benefit	We fully agree with the referee that a cell line sample does not represent the complexity of most biological samples. The authors have ongoing studies targeting PBMC subsets. Since a broad discussion of immunological PBMC subsets would – in our

greatly by the proof-of-concept sort and subsequent analyses of more complex samples, such as PBMCs (<https://www.biolegend.com/en-gb/products/veri-cells-pbmc-12260?GroupID=BLG2181> or other source of multiple cell types, like mouse spleens). Can this instrument **replace or complement conventional sorters by truly identify cells**, and accurately sort them, based on their light scattering properties? Can you **identify monocytes, B cells and T cells** based only on multiple angle scattering properties? Are T cells, for instance, producing **clusters with strong association to their phenotypical diversity?**

opinion – be beyond the scope of our manuscript, we selected a reduced subset and conducted experiments on the identification of B cells, as well as separation of CD4 against CD8 and CD45RA from CD45RO cells. These results were added to the manuscript in separate Sections.

Line 18: Furthermore, we demonstrate the ability to classify human peripheral blood T- and B-cell subsets.

Line 53-54: In addition, we have identified subgroups of T and B cells solely on the basis of the properties of their scattered light.

Line 80-91 (Sample preparation: PBMC samples)

Line 291-309 (Analysis of PBMC subsets), Figure 8 added:

	Line 356-361: As a proof-of-concept of further applications, we analyzed PBMC subsets only by means of scattered light and compared the results to conventional immunological staining approaches. Clusters of scattered light pulses with specificity for B cells, CD4 against CD8, as well as memory and naive T cells were observed. On the one hand, the analysis conditions were simplified due to a pre-selection of, e.g., only CD3 cells and there is no strict separation of the targeted cell subsets between the clusters. On the other hand, the authors are not aware of another label-free technique with similar capabilities. With further improvements to the data analysis such as artificial intelligence, we expected advances in the label-free identification of PBMC subsets.
3. Also, what is the function of the octagon array of PMTs? How many channels have been successfully tested with MAPS sorting? One huge limitation is the processing/computing power needed to resolve the clusters in the 10 microseconds needed between triggering and sorting. What is the data processing limit of the sort with 8 fluorescent channels? Can be circumvented by reducing the flow rate of the sample?	The function of the ‘Octagon’ is now described in the Methods Section of the manuscript. Line 112-113:). The ‘Octagon’ is a detector/ filter arrangement with up to eight PMTs coupled to a fiber. It allows to spectrally distribute the incoming light onto multiple detectors. The selection of the fibers/channels for sorting is non-trivial. The authors are working on an assessment for certain cell types including all available angles. However, the result will differ between cell types and an in-depth discussion is therefore not appropriate for this manuscript. We now comment on this issue in the Discussion Section. Line 369-370: Here, interference of the transmitted beam and the scattered light plays a major role and small deviations in the position can result in large differences in the obtained signals. The assignment of events to clusters is performed within 32 ns. Together with the wavelet transform, the sort decision can easily be made within the mentioned 10 μs. The available time between analysis (triggering) and sort decision is commonly larger than 100 μs, also in high-speed droplet sorters. Therefore, there is no need to reduce flow or sample rate. The data processing is rather limited by the number of digital-analog converter channels with appropriate sampling rate and digitization depth. Line 142-143: Thus, the time delay between a trigger event and the sort decision is typically below 10 μs and therefore much shorter than the travel time of several 100 μs from the laser interrogation point to the sort point.

4. Line 11-12: The known effect of sorting with the use of fluorescence is not backed up by any reference. This reviewer has successfully sorted cells for experiments requiring subsequent culture, many times. If the authors should provide a reference, data to benchmark survival of cells sorted with their cytometer (compared to a conventional one), or alternatively, remove such statement.	We accept your point and have addressed this concern by incorporating references that demonstrate the challenges associated with fluorescent labeling. In particular, as we wanted to avoid including references in the Abstract, we have included references 3 and 4 in the introductory part of the MS. Line 10-13: Flow cytometry is a key technology for the analysis and sorting of cells or particles at high throughput. Conventional and current flow cytometry is primarily based on fluorescent stains to detect the cells of interest. However, such stains also have disadvantages, as their effect on cells must be carefully tested to avoid effects on the results of the experiments. Alternative approaches using imaging or other label-free techniques often require highly sophisticated setups, are commonly limited in resolution, and produce challenging amounts of data. Line 25-27: However, their potential toxicity requires careful consideration and optimization to avoid adverse effects on cellular functions and experimental outcomes³. In addition, the staining process is becoming increasingly complex. This places high demands on sample preparation to avoid errors⁴.
5. Line 86: the shape of lenses, beams and long axis should be consistent in the diagram.	We understand and agree with the reviewer that the lens shape is not strictly correct. In a 2D schematic, however, the alternative would be to draw some lenses as rectangles and some slit apertures as solid lines. The authors doubt that this would improve the clarity and would prefer to stick with the current representation if the referee and the editor agree. The drawn beam reflects the beam shape in top view up to the point where it is overlaid with scattered light.
6. Line 88: what is the polarisation direction with respect with the sample flow at the interrogation point?	The authors apologize for unclear description. We specified the description and added the flow direction referring to the coordinate system in the schematic. Line 106-108: The long axis of the beam is parallel to the x-axis. After passing the periscope P, the cross section is rotated such that the long axis is parallel to y, the polarization direction at the interrogation point is along x. The sample flow direction is along y.
7. Figure 2: Please correct figure as it seems the direction of the flow and sort is upward.	The flow direction is indeed upwards due to the design of the mechanical sort unit.
8. Line 146: In the data availability statement authors should provide access to the Python script referenced (add link or ID).	The Python scripts are available in the linked repository. The code availability is described in a separate “Code Availability Statement” as required by the Journal guidelines. The repository was held in a closed-access state and is now publicly accessible.

9. The gating strategy for Figure 6 is incomplete. For clarity, the input sample should be shown together with the resulting 9 clusters sorted a-i. Where is the evidence of S, G1 and G2/M being defined over a bi-parametric distribution of singlets?	The authors apologize for the incomplete data representation. Since Fig. 3 already provides an overview of the sample, even though measured with a different device, we would prefer not to add more display items to the manuscript. In order to still provide all relevant information, a document with supporting information (SI) was created which contains the requested plots. Line 201: (cf. Figure S1) Line 254-255: The sorted samples were re-analyzed with a FACSCanto II flow cytometer and compared to an unsorted aliquot (see Figure S2).
Referee 2	
1. I have a feeling that the demonstration of distinguishing cell cycles was not greatly impressive but this technique is neat and it is interesting to see that lots of different information is averaged out in conventional forward scattering signals.	In response to this comment and the comments of referee 1, the authors conducted additional experiments demonstrating potential applications of the technology in label-free identification of PBMC subsets. These experiments are described and discussed in the manuscript now. See section 3.3 of the MS
2. In line 15, this description is not accurate “The custom-built flow cytometry setup comprises fiber arrays in forward and side scatter detection for angular resolution” because the authors show the multiangular detection in only forward direction. This was confusing and descriptions have to be more accurate in many places.	The authors apologize for this mistake and corrected the respective statement in the abstract. We also screened the manuscript for further potentially misleading descriptions and specified those. Line 14: and side Line 47: forward
3. In line 30, I understand that the concept of this technology has similarity or an analogy with ghost cytometry, especially in its label-free versions which integrate forward scattering waveforms (Ota et al, Science 2018, Ugawa et al, eLife 2021, Tsubouchi et al, Cell Rep Methods 2023) as feature extraction-free and image-quality-free high-content analysis methods. MAPS splits the optical signals in the frequency space, while ghost cytometry does it in the real image space. With references, it is constructive to address ghost cytometry and explain what is advantages and disadvantages	We greatly acknowledge this suggestion of the referee and added some more information in the respective parts of the Introduction and Discussion. However, the authors think that an in-depth comparison of different label-free technologies should be subject of a comprehensive review. Line 20-34: In order to make cell morphology accessible at high throughput in flow cytometry, imaging flow cytometers^{8,9} with sorting capabilities were developed recently. Furthermore, an approach based on structured illumination ready for cell sorting using AI data analysis was introduced^{10,11}. Although this approach does not use image reconstruction, it is constrained by the same limitations as other image-based methods¹² and has relatively low sorting speed.

	Line 375-377: Thus, the label-free approach presented here has the potential to overcome throughput limitations of other morphology-based approaches. Due to the use of separate detectors for different scattered light angles, the dynamic range is superior to a single camera.
4. In line 11, this statement can be more accurate: “State-of-the art flow cytometry relies on fluorescent stains to detect the cells of interest. (https://deepcell.com/, https://thinkcyte.com/ and so on)	The authors acknowledge that “state of the art” is misleading or ambiguous in this context. The authors wanted to stress that the majority of flow cytometry experiments are still fluorescence-based. We specified the sentence. The existence of alternative label-free techniques is mentioned in the abstract and further covered in the introduction. Line 10-11: Conventional and current flow cytometry is primarily based on fluorescent stains to detect the cells of interest. Line 30-32: In order to make cell morphology accessible at high throughput in flow cytometry, imaging flow cytometers^{8,9} with sorting capabilities were developed recently. Furthermore, an approach based on structured illumination ready for cell sorting using AI data analysis was introduced^{10,11}.
5. In line 201, it is interesting to see that the responses of channels 4 and 8 are so different. Can authors explain what is the possible reason fundamentally? If possible, can authors take an optical image of cell signals coming at a plane of “FCS away” in Figure 1? Is there no speckle effect? In addition, what is DWS in the legends of y-axis?	These two fibers are close to the rim of the transmitted beam on the FSC array. In this region, interference effects between transmitted and scattered light dominate the signal shapes. Already small deviations from central alignment can cause rather distinct changes in the signal pulse shapes. The authors appreciate the request and accordingly extended the description of the phenomenon in the Discussion Section of the manuscript. Line 369-370: Here, interference of the transmitted beam and the scattered light plays a major role and small deviations in the position can result in large differences in the obtained signals. An optical image of the cells cannot be taken with the current configuration since the image is at infinite distance. Speckle effects might be observed in capturing scattered light in all fibers at a certain time point. However, we would consider them as part of the scattering properties of complex biological objects such as cells. The authors apologize for the incomplete description of the axis label. We introduced the abbreviation “DWT” (discrete wavelet transform) now.

	Line 134: The discrete wavelet transform (DWT) is based on the algorithm of the PyWavelets package²⁷... Line 215: The discrete wavelet transform (DWT) coefficients...
6. In line 262, can I see the “gating for FSC-SSC intensities with a FACSAria II cell sorter”? I hope to see what happens if the authors apply a gate which is as good as possible in a reasonable manner.	The authors recognized that this might be helpful information for the readers as well. Therefore, we added plots for the gating strategy to the newly created Supporting Information document (Figure S3).
Further changes	
	Line 28: The... Line 362: proof of principle

REVIEWERS' COMMENTS:

Reviewer #1 (Remarks to the Author):

The authors have addressed most of my comments, however, point 7 is still a bit vague. Although the flow within the fluidics system might be upwards, the drop forms thanks to gravity, so its movement is always downward, unless I understood the system wrongly. In that case a better description of the fluidics and drop system should be provided to prevent any confusions of this sort.

Response to reviewers

Referee comment	Response by the authors
Referee 1	
1. The authors have addressed most of my comments, however, point 7 is still a bit vague. Although the flow within the fluidics system might be upwards, the drop forms thanks to gravity, so its movement is always downward, unless I understood the system wrongly. In that case a better description of the fluidics and drop system should be provided to prevent any confusions of this sort.	Thank you for your continued feedback on our manuscript. We apologize for any confusion regarding point 7. We would like to clarify that the sorting technology we use is based on a mechanical sort chamber, and no drops are formed in this system. As a result, the movement of the cells is indeed upwards. To improve the description and prevent any further confusion, we have included the following sentence in the manuscript: Line 144: When an event fulfills the sorting condition, the FPGA board sends a signal ramp to a piezo driver unit (APE, Berlin, Germany) to move the 'catcher tube' into the sort stream and deflect the object of interest into the sorting outlet. We hope this clarifies the mechanism and addresses your concern.